# Phenylalanine Ammonia Lyase *GmPAL1.1* Promotes Seed Vigor under High-Temperature and -Humidity Stress and Enhances Seed Germination under Salt and Drought Stress in Transgenic *Arabidopsis*

**DOI:** 10.3390/plants11233239

**Published:** 2022-11-25

**Authors:** Xi Zhang, Yingzi Shen, Kebing Mu, Wanhan Cai, Yangyang Zhao, Hang Shen, Xinhui Wang, Hao Ma

**Affiliations:** State Key Laboratory of Crop Genetics and Germplasm Enhancement, Nanjing Agricultural University, Nanjing 210095, China

**Keywords:** *GmPAL1.1*, seed vigor, high-temperature and high-humidity stress, salt stress, drought stress, ROS

## Abstract

Seed vigor is an important agronomic attribute, essentially associated with crop yield. High-temperature and humidity (HTH) stress directly affects seed development of plants, resulting in the decrease of seed vigor. Therefore, it is particularly important to discover HTH-tolerant genes related to seed vigor. Phenylalanine ammonia lyase (PAL, EC 4.3.1.24) is the first rate-limiting enzyme in the phenylpropanoid biosynthesis pathway and a key enzyme involved in plant growth and development and environmental adaptation. However, the biological function of PAL in seed vigor remains unknown. Here, *GmPAL1.1* was cloned from soybean, and its protein was located in the cytoplasm and cell membrane. *GmPAL1.1* was significantly induced by HTH stress in developing seeds. The overexpression of *GmPAL1.1* in *Arabidopsis* (OE) accumulated lower level of ROS in the developing seeds and in the leaves than the WT at the physiological maturity stage under HTH stress, and the activities of SOD, POD, and CAT and flavonoid contents were significantly increased, while MDA production was markedly reduced in the leaves of the OE lines than in those of the WT. The germination rate and viability of mature seeds of the OE lines harvested after HTH stress were higher than those of the WT. Compared to the control, the overexpression of *GmPAL1.1* in *Arabidopsis* enhanced the tolerance to salt and drought stresses during germination. Our results suggested the overexpression of *GmPAL1.1* in *Arabidopsis* promoted seed vigor at the physiological maturation period under HTH stress and increased the seeds’ tolerance to salt and drought during germination.

## 1. Introduction

Phenylalanine ammonia lyase (PAL, EC 4.3.1.24) catalyzes the deamination of phenylalanine to form trans-cinnamic acid. It is the entry point in the phenylpropanoid biosynthesis pathway and a rate-limiting enzyme that affects the synthesis of downstream products, such as flavonoids, anthocyanins, and lignin [1,2]. PAL has a significant effect on the growth and development and environmental adaptation in plant. Plants inhibited by the expression level of *PALs* showed sensitivity to UV-B light and had retarded growth and development, altered leaf and flower shape, and decreased pollen vigor [3,4]. PAL generally regulates plant biotic stress tolerance by changing the contents of salicylic acid and lignin, which were considered to have broad-spectrum resistance without affecting yield [5]. The up-regulated expression of *OsPALs* in rice increases the accumulation of salicylic acid and lignin biosynthesis, resulting in increased resistance to brown planthopper [6]. *PAL*-silenced wheat plants have lower aphid resistance [7]. *GmPAL2.1* in soybean improves disease resistance to *Phytophthora sojae* by increasing the salicylic acid and lignin content [8]. Furthermore, the role of PAL in plant abiotic stress response is mainly to reduce oxidative damage through the antioxidant pathway. The tolerance of wheat spikes under drought stress is stronger than that of flag leaves, which may be due to stronger phenylpropane metabolism pathways, such as the high expression and activity of TaPAL, TaC4H, and Ta4CL, as well as the high total content of phenolics and flavonoids, resulting in relatively high photosynthesis and low membrane damage [9]. UV-B-triggered NO production induced isoflavone synthesis and improved antioxidant capacity by upregulating the expression of key enzymes (such as PAL, chalcone isomerase, CHI, etc.) involved in phenylpropane metabolism [10].

Seed vigor is a complex agronomic trait, including seed longevity, germination speed, seedling growth, and early-stress tolerance [11]. Harsh terrestrial environments, which reduce seed germination, seed vigor, and crop yield during seed growth and development and seed storage, are a challenge for the seed industry [12,13]. In soybean, high-temperature and high-humidity (HTH) stress in summer during the soybean physiological maturity (R7) period would result in the reduced vigor of developing seeds [14]. The reason is that HTH stress affects the growth and development of seeds and causes damage to the leaves, thus preventing nutrients from being supplied to seeds [15]. Total polyphenol and total flavonoid contents, antioxidant activity, main polyphenol components, and PAL gene expression in seeds of buckwheat significantly have been found to increase during germination [16]. The contents of phenolic acids, lignans, and tocopherols in sesame seeds increased to varying degrees during 0–6 d of germination, and correspondingly, PAL was activated [17]. NaCl treatment in barley seed germination enhanced the gene expression and protein content of PAL, resulting in the enhancement of its activity [18].

Previous studies mainly focused on the changes of *PAL* expression level and enzyme activities during seed germination under normal and stress conditions [16,17,18]; however, the effect of *PAL* on seed viability and in response to HTH has not been well elucidated. In this study, the *GmPAL1.1* gene was isolated from a high-seed-vigor soybean cultivar, Xiangdou No. 3, and a low-seed-vigor cultivar, Ningzhen No. 1, and characterized, including its gene structure, phylogenetic relationship, and expression, in various tissues and in response to HTH in developing seeds. Moreover, its roles in affecting seed vigor formation under HTH stress and in response to salt and drought stresses during germination were explored. This study will allow us to understand the effect of PAL on seed vigor and in response to various abiotic stresses and provide a foundation for improving the tolerance to abiotic stresses in crops.

## 2. Results

### 2.1. Cloning and Subcellular Localization of GmPAL1.1

The coding sequences and promoter sequences of *GmPAL1.1* (GenBank accession No. NM_001357056) were cloned from soybean cultivars, Xiangdou No. 3 and Ningzhen No. 1, by the specific primers (Appendix A). The results showed that there was no difference in CDS and promoter sequences of *GmPAL1.1* between the two varieties, which were consistent with the reference sequence of soybean cultivar, Williams 82. Promoter analysis revealed that a variety of stress responsive cis-elements existed in promoter sequences of *GmPAL1.1*, including ABRE relative element, AC-elements, ARE, Box 4, CAAT-box, CCAAT-box, etc. (Appendix A). *GmPAL1.1* contained one intron and two exons (Appendix A). Its protein contained 712 amino acids and had a conserved domain of phenylalanine aminotransferase and multiple conserved sites for catalytic activity (Appendix A).

A phylogenetic tree was constructed to trace the evolutionary relationships among the PAL families in crops, such as soybean, *Arabidopsis*, maize, rape, and rice. Thirty PALs were selected, including eight from soybean, four from *Arabidopsis*, nine from rice, three from rape, and five from maize. The results showed these PALs were divided into two groups with GmPALs, AtPALs, and BnaPALs belonging to the same groups (Figure 1A). The GmPAL1.1–GFP fusion protein was constructed and transferred into the leaves of tobacco. The GFP protein was observed to be distributed in the cell membrane, cytoplasm, and nucleus, while the GmPAL1.1-GFP fusion protein was targeted to the cytoplasm and cell membrane (Figure 1B). The results indicated that GmPAL1.1 was located in the cytoplasm and cell membrane.

### 2.2. Expression Patterns of GmPAL1.1 in Various Tissues and under HTH Stress

The expression patterns of *GmPAL1.1* in different tissues and under HTH stress were detected by qRT-PCR. In the low-seed-vigor soybean cultivar, Ningzhen No. 1, *GmPAL1.1* was strongly expressed in the roots and leaves but weakly expressed in other tissues (Figure 2A). In the high-seed-vigor soybean cultivar, Xiangdou No. 3, the transcript level of *GmPAL1.1* was highest in the roots and leaves, followed by the stem and the seed at the R5 (beginning of seed development) stage (Figure 2B). Under the HTH stress, compared to the corresponding CK, *GmPAL1.1* was significantly up-regulated at 6 h, 96 h, and 168 h (*p* < 0.05 or 0.01) in the developing seeds in Ningzhen No. 1 at the physiological maturity (R7) stage (Figure 2C), while markedly (*p* < 0.01) up-regulated at 6 h and 168 h and significantly (*p* < 0.01) down-regulated at 48 h and 96 h in the developing seeds in Xiangdou No. 3 (Figure 2D).

### 2.3. The Overexpression of GmPAL1.1 in Arabidopsis Reduced ROS Accumulation in Plants at Physiological Maturity Stage under HTH Stress

The function of PAL has been widely studied in other plants, and it has been found to play important roles in plant growth and development and in response to stresses, but it has been rarely studied in seed vigor. Thus, we constructed a *GmPAL1.1* overexpression vector and transformed it into an *Arabidopsis* plant, and two *Arabidopsis* lines with the highest expression levels of *GmPAL1.1* (OE1, OE2) were selected for clarifying its functions in seed vigor (Appendix A). The WT and OE lines at the physiological maturity stage were treated under the normal condition (control) and HTH stress for 3d, and then, the seeds and leaves were obtained immediately and used for the determination of physiological indexes. The results showed the developing seeds of the OE lines accumulated a lower (*p* < 0.01) level of ROS than those of the WT, and the accumulation of H_2_O_2_ in the leaves in the OE lines was lower than in the WT after the HTH treatment (Figure 3A–C). After the HTH stress, the content of flavonoids in the leaves in the OE lines was significantly (*p* < 0.01) higher than in the WT; however, there was no marked (*p* > 0.05) difference in anthocyanins content in the leaves between the WT and OE lines (Figure 3D). Furthermore, the overexpression of *GmPAL1.1* was found to significantly (*p* < 0.01) increase the activities of SOD, POD, and CAT and reduce the production of MDA in the leaves (Figure 3D). All these results indicate that the overexpression of *GmPAL1.1* in *Arabidopsis* reduced ROS accumulation in developing seeds and in leaves at the physiological maturity stage under HTH stress, and the ROS accumulation was reduced in leaves by increasing the flavonoid content and the activities of SOD, POD, and CAT.

### 2.4. The Overexpression of GmPAL1.1 in Arabidopsis Enhanced Tolerance of Developing Seeds to HTH Stress and Produced High-Vigor Seeds

The seeds from the treated WT and OE lines under the normal condition (control) and HTH stress at the physiological maturity stage, respectively, were harvested at the full-maturity stage, and their germination performance was investigated. For the normal condition (control) treatment, there was no significant difference in seed germination rate and viability between the OE lines and WT (Figure 4A,C,D). However, for the HTH stress, the seed germination rate and viability of both the OE lines and WT decreased, but the OE lines displayed higher germination rates and viability than the WT (Figure 4B–D). Our results indicated that the OE lines had stronger tolerance to HTH stress than the WT at the physiological maturity stage, and the overexpression of *GmPAL1.1* in *Arabidopsis* could enhance the tolerance of developing seeds to HTH stress, thus producing higher vigor seeds.

### 2.5. The Overexpression of GmPAL1.1 in Arabidopsis Improved Seed Vigor and Enhanced Tolerance to Salt and Drought Stresses during Germination

To further verify the function of *GmPAL1.1* in improving seed vigor, germination performance of the mature seeds harvested from the normal-grown WT and OE lines was investigated under salt and drought stresses. TTC staining indicated that there was no significant (*p* > 0.05) difference in seed viability between the OE lines and WT under the control conditions. However, under the salt and drought stresses, the OE lines maintained a higher (*p* < 0.01) degree of seed staining than the WT (Figure 5A,B). Moreover, DAB staining demonstrated that the WT seeds accumulated higher (*p* < 0.01) concentration of H_2_O_2_ than the OE lines (Figure 5C,D), indicating the overexpression of *GmPAL1.1* reduced the ROS accumulation in seeds during germination under salt and drought stresses.

Moreover, the seeds of the WT and OE lines were exposed to 150 mM NaCl and 250 mM mannitol, respectively, and then, their germination rates and root lengths were investigated. No significant differences (*p* > 0.05) were found in the germination rates and root lengths between the OE lines and WT under the control condition. However, under the salt and drought treatments, the germination rates and root lengths of the OE lines were higher than those of the WT (Figure 6).

Taken together, all these results indicated that the OE lines have higher seed vigor than the WT, implying that the overexpression of *GmPAL1.1* in *Arabidopsis* could improve seed vigor and enhance the tolerance to salt and drought stresses.

## 3. Discussion

Evolution has been induced by the environment for plants to adapt to harsh terrestrial environments since the emergence of plants, and the phenylpropanoid biosynthesis pathway is the adaptive performance of terrestrial plants evolved to cope with various stresses [19]. PAL affects the synthesis of diverse secondary metabolites in the phenylpropanoid biosynthesis pathway [4]. It fulfills numerous physiological functions and is essential for plant growth, development, and plant–environment interactions [20]. For example, the expression of *PAL* is stimulated during growth and development by a variety of stresses, including attacks, wounding, UV irradiation, low temperatures, and salt [18,20]. In the present study, a PAL gene, *GmPAL1.1*, was cloned in soybean. *GmPAL1.1* contained one intron and two exons (Appendix A). Its protein contained 712 amino acids, had a conserved domain of phenylalanine aminotransferase and multiple conserved sites for catalytic activity, and was located in the cytoplasm and cell membrane (Figure 1B, Appendix A). Expression analysis indicated that it was strongly expressed in the roots and leaves in two varieties, which are consistent with those of previous studies [8,21]. Interestingly, the transcript level of *GmPAL1.1* in the developing seed at the R5 stage was only inferior to that in the root in Xiangdou No. 3, while lowest in Ningzhen No. 1 (Figure 2A,B). Moreover, under the HTH stress, *GmPAL1.1* in developing seeds was significantly up-regulated at 6 h and 168 h, but showed the opposite expression model at 96 h in the two soybean varieties (Figure 2C,D). All these indicated that *GmPAL1.1* might adopt different expressive patterns during seed development and under HTH stress among the soybean varieties with different vigor. All the results indicated *GmPAL1.1* might be involved in the development of seeds and their response to HTH stress in soybean.

Numerous studies have investigated the promoters of *PAL* genes. The nuclear factors have been found to probably recognize AT-rich sequence motifs and activate the gene encoding PAL by a fungal elicitor [22]. The *PAL1* promoter activities are associated with plant development and are inducible by various stress agents, while the *PAL2* promoter activities are associated with the structural development of the plant and its organs [23]. *Cis*-elements in gene promoters play a key role in gene expression and environmental regulation [24]. To better understand its function, 2.0 kb non-coding sequences upstream of the *GmPAL1.1* translation start site (TSS) were adopted to predict the *cis*-regulatory elements. The promoter region of *GmPAL1.1* was found to possess multiple motifs. Among them, CAAT-box, CCAAT-box, and G-Box were light response elements, while STRE (stress response element under heat stress), MYB, MYC, W-box (WRKY binding site), ABRE (cis-acting element of abscisic acid-response), ERE (ethylene-response element), and GARE motif (gibberellin-responsive element) participated in responses to abiotic stresses and plant hormones (Appendix A). In addition, AC-elements, which are MYB-binding elements and specific elements of the genes of the phenylpropanoid biosynthesis pathway and are related to the synthesis of downstream metabolites, existed in the *GmPAL1.1* promoter (Appendix A). Interestingly, there was no difference in the promoter sequence and coding sequence of *GmPAL1.1* between the two cultivars, Xiangdou No. 3 and Ningzhen No. 1. The analysis of the *cis*-regulatory elements in the promoter suggested *GmPAL1.1* is involved in the regulation of plant growth and development and responses to multiple abiotic stresses and exogenous hormones.

When seeds respond to unfortunate environmental conditions, a high level of ROS is induced, which then leads to consequent oxidative damage and acts as a stressor that significantly inhibits seed vigor under stress conditions [24,25,26]. It is well-known that flavonoids and antioxidant enzymes are effective scavengers of ROS in plant response to various stresses [27]. Therefore, PAL, SOD, POD, and CAT are often used as enzymes of the antioxidant system to evaluate the roles of genes in stresses [28]. In the present study, H_2_DCFDA and DAB were used to detect ROS levels, and it was found that the developing seeds and the leaves of the *GmPAL1.1* overexpression *Arabidopsis* lines had lower ROS accumulation than those of the WT at the physiological maturity period under HTH stress (Figure 3A–C). Subsequent studies demonstrated that the OE lines had higher flavonoid content, higher antioxidant enzyme activities (SOD, POD, and CAT), and lower MDA content in leaves than the WT at the physiological maturity period under HTH stress (Figure 3D). The wild *Arabidopsis* produced wrinkled seeds under HTH stress at the physiological maturity period, while *GmPAL1.1* overexpressing *Arabidopsis* had more plump seeds (Appendix A). Furthermore, the germination rates and TTC staining of the seeds harvested at the full-maturity stage from the WT and OE lines treated with HTH stress at the physiological maturity period were investigated. The results showed the seed germination rates and viability of both the OE and WT decreased, but the OE lines displayed higher germination rate and viability than the WT after HTH stress (Figure 4A–D). All these results implied that the overexpression of *GmPAL1.1* in *Arabidopsis* improved the seed vigor under HTH stress and the plant tolerance to HTH stress through reducing the hazards of ROS. However, the roles of the PAL pathway and the antioxidant enzyme pathway in enhancing antioxidation of developing seeds and leaves by the overexpression of *GmPAL1.1* in *Arabidopsis* were unknown. Notably, BplMYB46 was found to be able to bind the TC-box, GT-box, and E-box motifs in the promoters of PAL, POD, and SOD genes, which function in abiotic stress tolerance [29]. The results indicated there was some relationship in regulation between the PAL pathway and the antioxidant enzyme pathway. Therefore, it is worth further investigation on the relationship in regulation between the PAL pathway and antioxidant enzyme pathway in response to abiotic stresses. 

Abscisic acid (ABA) is an extremely important regulator of seed dormancy and germination. Genotype and environment influence the biosynthesis of ABA in seed, inducing differing depths of primary dormancy during seed development [11]. There are few studies on the function of *PALs* in seed dormancy. Our results indicated that overexpression of *GmPAL1.1* reduced the seed germination rate under ABA treatment, implying that *GmPAL1.1* increased ABA sensitivity and seed dormancy in *Arabidopsis* seeds (Appendix A). *GmPAL1.1* might be a positive regulator of seed dormancy. Our subsequent study will focus on whether it regulates seed dormancy by affecting the ABA regulatory pathway under HTH stress. 

High-vigor seeds should have a better ability to complete germination in variable environmental conditions [30,31]. In the present study, the mature seeds were harvested from the WT and OE lines grown under normal condition and their germination capacity was evaluated under salt and drought stresses. The overexpression of *GmPAL1.1* in *Arabidopsis* was found to enhance the tolerance to salt and drought stresses during germination, through the reduction of H_2_O_2_ concentration (Figure 5 and Figure 6, Appendix A). All these results indicated that *GmPAL1.1* might be an elite gene to improve seed vigor and tolerance in wide environmental stress conditions. 

## 4. Materials and Methods

### 4.1. Plant Materials and Treatments

Soybean cultivars, Ningzhen No. 1 (a cultivar with low seed vigor) and Xiangdou No. 3 (a cultivar wit high seed vigor), were used in this study [14]. Plants were grown under normal conditions. For the tissue expression assay, roots, stems, and leaves were collected at the V2 (second trifoliate leaf unrolls) stage, flowers were collected at the R2 (full bloom) stage, R5-seeds/pods and R7-seeds/pods were collected at the R5 (beginning of seed development) and R7 (physiological maturity) periods. For the expression assay under HTH treatment, plants were grown to the R7 stage under normal condition and then stressed under HTH (40 °C/24 °C, 98% RH/70% RH, light for 16 h/dark for 8 h) condition for 7 d. During the HTH stress, the developing seed samples were obtained at 0, 6, 12, 24, 48, 96, and 168 h, respectively, and stored at −80 °C after being quick-frozen in liquid nitrogen for subsequent quantitative analysis.

The surface-sterilized seeds of wild type (WT) and transgenic *Arabidopsis* lines were planted on 1/2 MS medium. The seven-day-old transgenic and wild-type seedlings were transferred to pots and grew under normal condition (26 °C/24 °C, RH 70%, light for 16 h/dark for 8 h) to the physiological maturity stage. The HTH treatments of the WT and OE lines were conducted in an artificial aging equipment (40 °C/24 °C, 98% RH/70% RH, light for 16 h/dark for 8 h) at the physiological maturity stage for 3 d. Then, all the plants grew under normal conditions, and seeds were harvested at seed full-maturity stage for further experiments.

The surface-sterilized *Arabidopsis* seeds of wild type and transgenic lines were planted on 1/2 MS media (control) with or without 150 mM NaCl and 250 mM mannitol, respectively, to detect seed germination. The germination rates were investigated on the seventh day. For the root length assay, the sterilized seeds of transgenic lines and WT were grown on 1/2 MS media for 7 d, and then, the seedlings were transplanted on 150 mM NaCl and 250 mM mannitol conditions, respectively, for 5 d to examine the root length.

### 4.2. Cloning and Bioinformatics Analysis of GmPAL1.1

Soybean cultivars, Xiangdou No. 3 and Ningzhen No. 1, were used to clone the gene of *GmPAL1.1*. The cDNA sequences of *GmPAL1.1* and other species were from NCBI (https://www.ncbi.nlm.nih.gov/) (accessed on 11 October 2020). ClusterX2 is used for sequence alignment. SMART (http://smart.embl-heidelberg.de/) (accessed on 11 October 2020) was used for protein functional domain prediction. Phylogenetic trees were constructed using MEGA-X, based on the adjacency (NJ) method. Plant CARE (http://bioinformatics.psb.ugent.be/webtools/plantcare/html/) (accessed on 11 October 2020) was used to analyze the promoter sequences.

### 4.3. Subcellular Localization

The ORF of *GmPAL1.1* was constructed into pBinGFP4 vector. The vector was introduced into tobacco leaves by the *Agrobacterium tumefaciens*-mediated method. The transformed tobacco leaves were cultured in dark for 24 h, grown in light for 2–3 d, and then, observed by a laser confocal microscope (Zeiss LSM780, Jena, Germany) [32].

### 4.4. Expression Analysis

Total RNA was extracted from soybean or *Arabidopsis* using TRIeasy^TM^ Total RNA Extraction Reagent (YEASEN, Shanghai, China). Reversing transcription for the first strand cDNA synthesis was performed using a ToloScript RT EasyMix for qPCR (TOLOBIO, Shanghai, China). The qRT-PCR analysis was performed on CFX96 real-time PCR detection (CFX96, Bio-Rad, Waltham, MA, USA) systems with SYBR Green Real-Time PCR Master Mix (TaKaRa, Dalian, China). The primers were designed by Premier 5.0 in which the soybean actin gene was used as a control. The quantitative variations of gene expression between the examined replicates were evaluated by the 2^−ΔΔCt^ method described previously [33]. Three independent biological repeats were performed to ensure accurate statistical analysis. The specific primers are listed in Appendix A.

### 4.5. Genetic Transformation and Screening of Transgenic Lines in Arabidopsis

To obtain overexpressing *Arabidopsis* plants, the ORF region of *GmPAL1.1* was constructed into the pRI101 vector. Subsequently, transformation into *Arabidopsis* was performed using the *Agrobacterium tumefaciens*-mediated floral dip method [33]. Positive transgenic lines were selected using MS medium containing kanamycin (50 mg/L). The expression level of *GmPAL1.1* was analyzed by qRT-PCR. Two *Arabidopsis* lines overexpressing *GmPAL1.1* were selected. Primers are shown in Appendix A.

### 4.6. TTC-Staining, DAB-Staining, and ROS Content Assays

TTC-staining assay was used to show seed viability. Seeds were stained with 1% TTC solution and then kept in the incubator under dark conditions at 30 °C for 1 d. These staining seeds were observed using a MVX10 stereo fluorescence microscope (Olympus, Tokyo, Japan) [34].

DAB-staining assay was used to show H_2_O_2_ content. The leaves were incubated overnight in DAB-staining solution (1.0 mg/mL DAB in 50 mM Tris-HCl buffer, pH 5.0) at 25 °C in dark. The leaves and seeds were photographed using a VX10 stereo fluorescence microscope (Olympus, Tokyo, Japan) [35].

H_2_DCFDA-staining assay was used to show the ROS level. Seeds were transferred to H_2_DCFDA Buffer (1 mM H_2_DCFDA, 10 mM Hepes-NaOH, pH 5.7) and incubated in the dark for 30 min. It was observed by a laser confocal microscope (Zeiss LSM780, Jena, Germany) at 488 nm excitation and at 525 nm emission [36]. Each sample had at least three biological replicates.

### 4.7. Measurements of Flavonoids and Anthocyanins Contents

Determination of total flavonoids was conducted, as previously described [37,38]. Leaves were ground in liquid nitrogen and extracted at 4 °C for 24 h in 80% methanol, then reacted at 72 °C for 40 min in 1% HCl. The supernatant was collected by centrifugation. Flavonoid content was measured spectrophotometrically at 270 nm. Each sample had at least three biological replicates.

Determination of anthocyanin content was conducted, as previously described [38]. Leaves were ground in liquid nitrogen and extracted in 1% HCl-methanol at 4 °C for 24 h, and then added to chloroform. The supernatant was collected by centrifugation, and the anthocyanin content was measured spectrophotometrically at 525 nm. Each sample had at least three biological replicates.

### 4.8. Measurements of SOD, POD, CAT, and MDA Contents

The determination steps of antioxidant enzyme activity and MDA content of *Arabidopsis thaliana* leaves were conducted, as previously described [36]. In brief, the activity of superoxide dismutase (SOD) was determined by a nitrogen blue tetrazole method, while the activity of peroxidase (POD) was determined by a guaiacol method. Catalase (CAT) activity was determined by monitoring the decrease in absorbance at 240 nm due to the decomposition of H_2_O_2_ [39]. The content of MDA was measured using a thiobarbituric acid-based method. Each sample had at least three biological replicates.

### 4.9. Statistics Analysis

All statistical analysis was conducted by GraphPad Prism 9.0 (San Diego, CA, USA), and comparisons were made by means of the analysis of variances followed by Student’s *t* test. The statistical significance was considered as *p* < 0.05 or 0.01.

## 5. Conclusions

The PAL family gene *GmPAL1.1* in soybean contained one intron and two exons, and its protein contained 712 amino acids. *GmPAL1.1* was significantly induced by HTH stress. The overexpression of *GmPAL1.1* in *Arabidopsis* enhanced the tolerance of developing seeds to HTH stress, thus producing higher vigor seeds and enhanced the tolerance to salt and drought stresses during germination. Moreover, the overexpression increased flavonoid content and the activities of SOD, POD, and CAT in leaves at physiological maturity stage.

## Figures and Tables

**Figure 1 plants-11-03239-f001:**
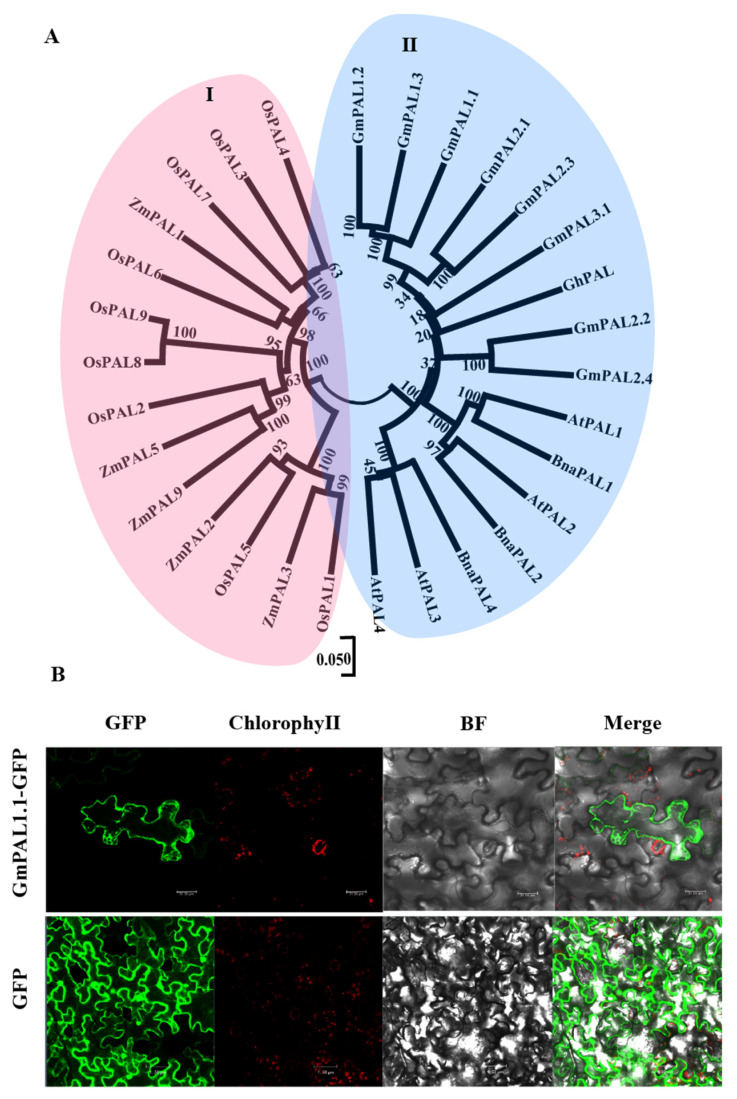
Sequence analysis and subcellular localization. (**A**) Phylogenetic analysis of PALs from soybean and other crops. Phylogenetic tree was constructed by neighbor-joining method using MEGA-X software. The amino acid sequences of PALs were all obtained from the NCBI database. Gm, *Glycine max*; At, *Arabidopsis thaliana*; Bna, *Brassica napus*; Os, *Oryza sativa*; Zm, *Zea mays*. (**B**) Subcellular localization of GmPAL1.1 in tobacco. The coding sequence of *GmPAL1.1* without stop codon was constructed into pBinGFP4-GFP vector and transformed into four-week-old tobacco leaves using *Agrobacterium*-mediated transformation. The injected area was observed using laser confocal microscope. GFP, empty carrier protein; GmPAL1.1-GFP, fusion protein; BF, brightfield; Chlorophy II, autofluorescence emitted by chloroplasts.

**Figure 2 plants-11-03239-f002:**
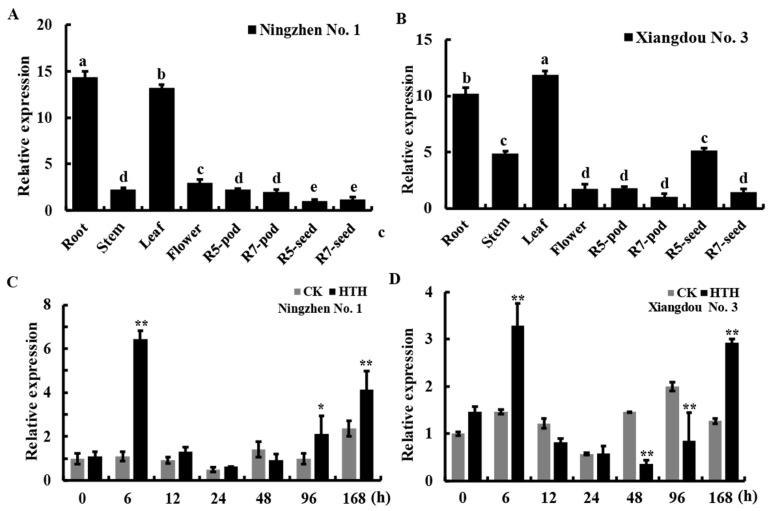
Expression analysis of *GmPAL1.1*. (**A**,**B**) Relative expression of *GmPAL1.1* in different tissues of soybean cultivars, Ningzhen No. 1 and Xiangdou No. 3. Different tissue samples of soybean grown under the same growth conditions were obtained for qRT-PCR analysis. (**C**,**D**) Relative expression levels in developing seeds of *GmPAL1.1* in Xiangdou No. 3 and Ningzhen No. 1 under HTH stress. Soybean plants grown to physiological maturity (R7) stage were treated with HTH stress (40 °C/24 °C, 98% RH/70% RH, light for 16 h/dark for 8 h) for 7 d, and during the treatment, samples of developing seeds were obtained at 0, 6, 12, 24, 48, 96, and 168 h, respectively. The control (CK) plants were grown under normal condition. Values are mean ± SD of three independent biological replicates. The means with different lowercases indicated significant differences at 5% probability level. * & ** indicated significant differences at 5% and 1% probability levels between treatment and corresponding CK, respectively.

**Figure 3 plants-11-03239-f003:**
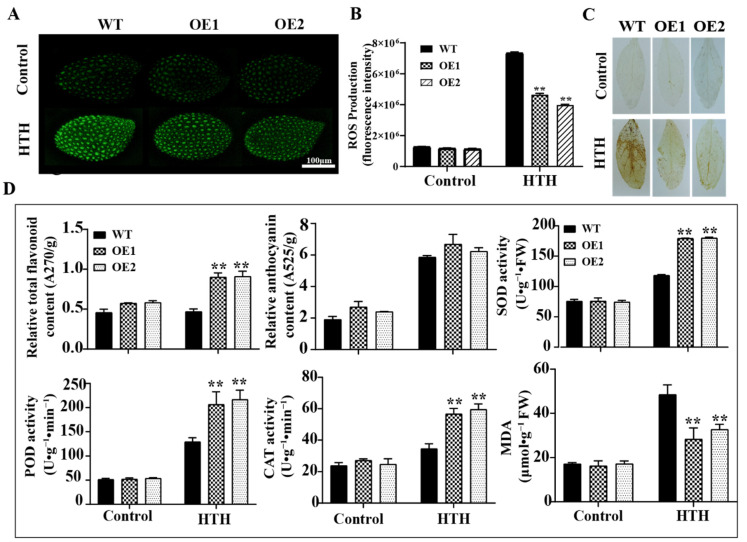
ROS content and physiological indexes of *GmPAL1.1* overexpressing *Arabidopsis* (OE) lines and WT at physiological maturity stage after HTH stress. Control plants (WT and OE lines) were placed in a normal growth chamber (26 °C/24 °C, RH 70%, light for 16 h/dark for 8 h) for 3 d, and HTH-stressed plants (WT and OE lines) were put into artificial aging equipment (40 °C/24 °C, 98% RH/70% RH, light for 16 h/dark for 8 h) for 3 d. (**A**,**B**) ROS content in developing seeds. Developing seeds of *Arabidopsis thaliana* were incubated for 30 min in H_2_DCFDA buffer (10 mM H_2_DCFDA, 10 mM Hepes-NaOH, pH 5.7), photographed and observed using a laser confocal microscope, and the fluorescence intensity was measured by ImageJ. (**C**) DAB staining of leaves. (**D**) Flavonoids and anthocyanins content, antioxidant enzyme activities, and MDA content in leaves. Values are mean ± SD of three independent biological replicates, ** indicated significant differences at 1% probability levels between WT and transgenic *Arabidopsis* lines under HTH treatments.

**Figure 4 plants-11-03239-f004:**
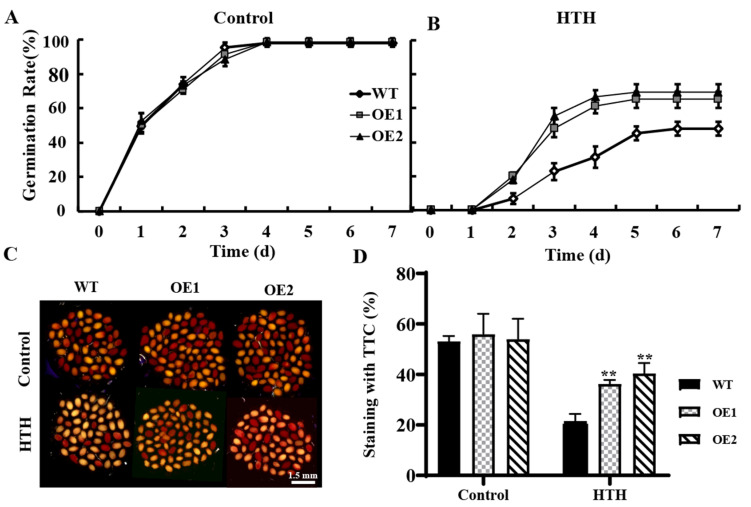
Germination rates and viability of the mature seeds harvested from the treated *GmPAL1.1* overexpressing *Arabidopsis* (OE) lines and WT under normal condition and HTH stress at physiological maturity stage. Control plants (WT and OE lines) were placed in a normal growth chamber (26 °C/24 °C, RH 70%, light for 16 h/dark for 8 h) for 3 d. HTH-stressed plants (WT and OE lines) were put into artificial aging equipment (40 °C/24 °C, 98% RH/70% RH, light for 16 h/dark for 8 h) for 3 d. (**A**) Seed germination rates of WT and OE lines after the control treatment. (**B**) Seed germination rates of WT and OE lines after the HTH stress. (**C**,**D**) TTC staining of WT and OE lines after the control treatment and HTH stress, respectively. TTC staining was used to evaluate seed viability. Red seeds indicate viable seeds. Values are mean ± SD of three independent biological replicates. ** indicated significant differences at 1% probability levels between WT and transgenic *Arabidopsis* lines.

**Figure 5 plants-11-03239-f005:**
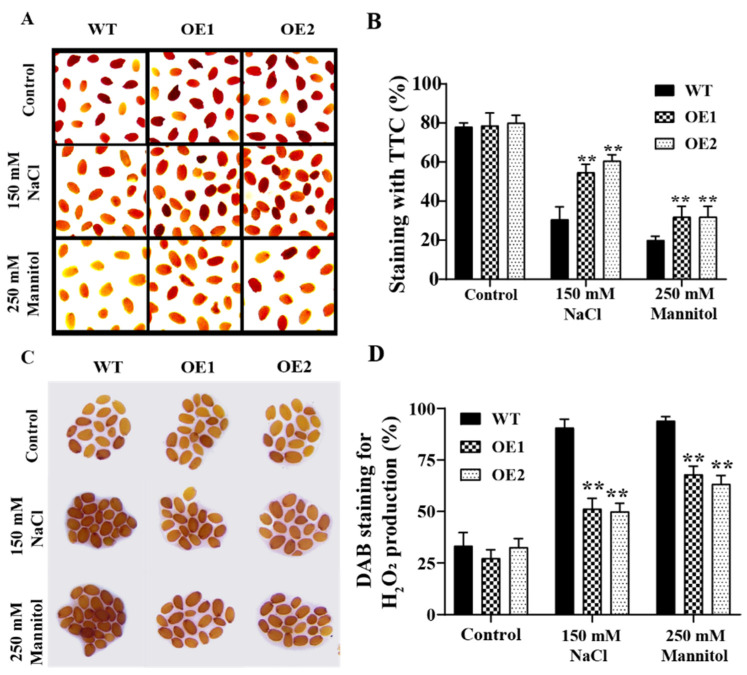
TTC staining and DAB staining of the mature seeds harvested from the *GmPAL1.1* overexpressing *Arabidopsis* (OE) lines and WT grown under normal condition. (**A**,**B**) Seed viability of the WT and OE lines germinated under 150 mM NaCl and 250 mM mannitol conditions for 12 h, respectively. Dark-red staining indicated viable seeds, while light-pink staining indicated reduced seed viability. (**C**,**D**) DAB staining of the WT and OE lines of seeds germinated under 150 mM NaCl and 250 mM mannitol conditions for 12 h, respectively. Values are mean ± SD of three biological replicates. ** indicated significant differences at 1% probability levels between the WT and transgenic *Arabidopsis* lines.

**Figure 6 plants-11-03239-f006:**
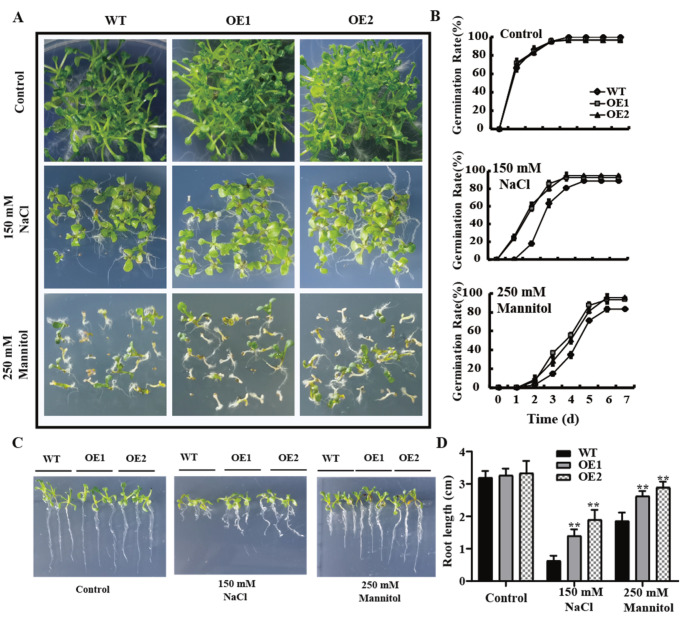
Germination rates and root lengths of the mature seeds harvested from the *GmPAL1.1* overexpressing *Arabidopsis* (OE) lines and WT grown under normal condition. (**A**) Germination performance of the WT and OE lines under salt and drought stresses. The seeds of the WT and OE lines were grown for 7 d under 150 mM NaCl and 250 mM mannitol conditions, respectively. (**B**) Germination rates of WT and OE lines under salt and drought stresses. The germination rates were recorded on the seventh day. Each experiment was repeated three times with each replicate containing 50 seeds. (**C**,**D**) Root lengths of WT and OE lines under salt and drought stresses. The sterilized seeds of the OE lines and WT were grown on 1/2 MS media for 7 d, and then, the seedlings with the same root lengths and growth states were selected to grow for 5 d under salt and drought treatments. Values are mean ± SD of three biological replicates. ** indicated significant differences at 1% probability level between WT and transgenic *Arabidopsis* lines.

## Data Availability

Not applicable.

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
