# Peer review of "Phenylalanine Ammonia Lyase GmPAL1.1 Promotes Seed Vigor under High-Temperature and -Humidity Stress and Enhances Seed Germination under Salt and Drought Stress in Transgenic Arabidopsis"

_plants, 2022, doi:10.3390/plants11233239_

Round 1
Reviewer 1 Report
The manuscript "Phenylalanine ammonia lyase GmPAL1.1 promotes seed vigor under high temperature and humidity stress and enhances seed germination under salt and drought stress in transgenic Arabidopsis" gives a talk about the GmPAL1.1 regulates the seed vigor under high temperature and humidity (HTH) stress in transgenic Arabidopsis. This study investigated the expression patterns of GmPAL1.1, and then transformed the gene in soybean into Arabidopsis to identify the gene functions under high temperature and high humidity stress, salt and drought stresses. The research is interesting and essential for the study of seed vigor. However, there are still some problems contained in the manuscript. I have some comments as follows, and hope these are useful in improving the manuscript.
1. The authors found that GmPAL1.1 overexpression reduced oxidative damage and produced higher vigor seeds in Arabidopsis. Were there morphological changes in seeds harvested after HTH stress? Please specify.
2. Since seed stress treatment and seed harvest are not in the same growth period, does GmPAL1.1 affect seed development or seed vigor formation in Arabidopsis during the period between after HTH stress and before seed harvest? Please specify.
3. The authors transferred GmPAL1.1 into Arabidopsis for functional studies with the ultimate aim of investigating the gene's influence on seed vigor formation in soybean seeds under HTH stress. Currently, the stable genetic transformation has been achieved in soybean. Why did not the author overexpress the gene in soybean?
Some errors:
1. Lines 64-67 "a high-seed-vigor soybean cultivar Xiangdou No. 3 and a low-seed-vigor cultivar Ningzhen No. 1" -lack of the reference. Please add references.
2. Line 352, 'Leaves' instead of 'leaves'? typo
3. Line 362, 'were' instead of 'was'? typo
Author Response
Dear Reviewer: Thank you for your valuable and thoughtful comments and suggestions for our manuscript. We appreciate your excellent comments and suggestions. We have considered all of these suggestions and comments in revising our manuscript. The following lists our responses to your comments. Best Wishes, Hao Ma State Key Lab of Crop Genetics and Germplasm Enhancement, Nanjing Agricultural University, Nanjing, Jiangsu Province 210095, China Comments and Suggestions for Authors The manuscript "Phenylalanine ammonia lyase GmPAL1.1 promotes seed vigor under high temperature and humidity stress and enhances seed germination under salt and drought stress in transgenic Arabidopsis" gives a talk about the GmPAL1.1 regulates the seed vigor under high temperature and humidity (HTH) stress in transgenic Arabidopsis. This study investigated the expression patterns of GmPAL1.1, and then transformed the gene in soybean into Arabidopsis to identify the gene functions under high temperature and high humidity stress, salt and drought stresses. The research is interesting and essential for the study of seed vigor. However, there are still some problems contained in the manuscript. I have some comments as follows, and hope these are useful in improving the manuscript. 1. The authors found that GmPAL1.1 overexpression reduced oxidative damage and produced higher vigor seeds in Arabidopsis. Were there morphological changes in seeds harvested after HTH stress? Please specify. Author’s Response: Thanks for the comment. We observed and photographed the seeds after HTH stress. The results have been provided in Discussion section of revised manuscript as well as Supplementary Figure S3. 2. Since seed stress treatment and seed harvest are not in the same growth period, does GmPAL1.1 affect seed development or seed vigor formation in Arabidopsis during the period between after HTH stress and before seed harvest? Please specify. Author’s Response: GmPAL1.1 overexpression may continuously affect seed development as well as response to stress tolerance. We appreciate this comment and will focus on this issue later. 3. The authors transferred GmPAL1.1 into Arabidopsis for functional studies with the ultimate aim of investigating the gene's influence on seed vigor formation in soybean seeds under HTH stress. Currently, the stable genetic transformation has been achieved in soybean. Why did not the author overexpress the gene in soybean? Author’s Response: Thanks for your good suggestion. The transgenic soybean work is ongoing but will take longer time. Later, we will publish the work of transgenic soybean in future. Some errors: 1. Lines 64-67 "a high-seed-vigor soybean cultivar Xiangdou No. 3 and a low-seed-vigor cultivar Ningzhen No. 1" -lack of the reference. Please add references. Author’s Response: The reference has been added to Materials and Methods in the revised manuscript as per your suggestions. 2. Line 352, 'Leaves' instead of 'leaves'? typo Author’s Response: It has been corrected in the revised manuscript as per your suggestions. 3. Line 362, 'were' instead of 'was'? typo Author’s Response: It has been corrected in the revised manuscript as per your suggestions.Reviewer 2 Report
Does the introduction provide sufficient background and include all relevant references?
The Introduction section of the manuscript provides the theoretical background of the research.
Is the research design appropriate?
This research design is appropriate for the Journal's profile.
Are the methods adequately described?
The section Materials and Methods is described in a complete, detailed, and simple way. Only note: the The Materials and Methods section precedes Coclusion section and follows the Results and Conclusions sections. This provision of the paragraphs complicates the understanding of the scientific article..
Are the results clearly presented?
In the Results section, the analysis results are described in a prominent, concise, and precise manner. In addition, even the figures help significantly understand the results.
Are the conclusions supported by the results?
The manuscript provides essential conclusions and implications in the section “Discussions.”
The conclusions are supported by the results, which are widely discussed quantitatively and qualitatively.

Author Response
ThanksReviewer 3 Report
1.Figure 2. Expression analysis
At R5 stage, relative expression level of GmPAL was obvious between Xiangdou and Ningzhen under HTH stress, explaining the possible reasons.
2, In Figure 2. Where is C and D in the annotation? Please give a detailed explanation of the expression difference between the two varieties at 96h.
3, Whether PAL gene is related to seed dormancy will be discussed in another section.
Author Response
Dear Reviewer: Thank you for your valuable and thoughtful comments and suggestions for our manuscript. We appreciate your excellent comments and suggestions. We have considered all of these suggestions and comments in revising our manuscript. The following lists our responses to your comments. Best Wishes, Hao Ma State Key Lab of Crop Genetics and Germplasm Enhancement, Nanjing Agricultural University, Nanjing, Jiangsu Province 210095, China Comments and Suggestions for Authors 1.Figure 2. Expression analysis At R5 stage, relative expression level of GmPAL was obvious between Xiangdou and Ningzhen under HTH stress, explaining the possible reasons. Author’s Response: We appreciate your comment. We added the discussion in the Discussion. 2, In Figure 2. Where is C and D in the annotation? Please give a detailed explanation of the expression difference between the two varieties at 96h. Author’s Response: It has been corrected in the revised manuscript. We added the discussion about the different expression between the two varieties in the Discussion. 3, Whether PAL gene is related to seed dormancy will be discussed in another section. Author’s Response: We appreciate your comment. We have added the discussion on the relationship between GmPAL1.1 and seed dormancy as well as Supplementary Figure S4.